# Caffeic Acid Inhibits Degranulation, Cytokine Secretions, and *IP3 Receptor 2* Gene Expression in Compound 48/80-Stimulated Mouse P815 Mast Cells

**DOI:** 10.3390/ijms262010086

**Published:** 2025-10-16

**Authors:** Kai-Chi Chang, Jin-Yuarn Lin

**Affiliations:** Department of Food Science and Biotechnology, National Chung Hsing University, 250 Kuo-Kuang Road, Taichung 40227, Taiwan; ninainfocus@gmail.com

**Keywords:** allergic inflammation, caffeic acid, *inositol 1,4,5-triphosphate receptor 2*, mast cell degranulation, mouse P815 mast cells

## Abstract

Caffeic acid (CA), 3-*O*-caffeoylquinic acid (3-CQA), and 5-*O*-caffeoylquinic acid (5-CQA) were subjected to treating stimulated mouse P815 mast cells to unravel their antiallergic potential. β-Hexosaminidase release, appearance, morphology change, cytokine secretions, and degranulation-related pathway gene expressions, including *Mas-related G protein-coupled receptor, member B2* (*MRGP receptor B2*), and *inositol 1,4,5-triphosphate receptor 2* (*IP3 receptor 2*), in the stimulated mast cells were measured. An ELISA was used to determine the secreted cytokines. The relative gene expression folds were analyzed with reverse transcription real-time quantitative polymerase chain reaction. Correlations between gene expressions and different parameters were analyzed using the Pearson product–moment correlation coefficient (r). The results showed that CA had a superior effect than 3-CQA and 5-CQA on reducing β-hexosaminidase release, IL-4, and IL-6 cytokine secretions by the compound 48/80 (C48/80)- and 5-hydroxymethyl-2-furaldehyde (5-HMF)-stimulated mast cells. CA increased intact mast cell numbers but reduced granule releases, evidencing that CA may soothe activated mast cells. CA reduced *IP3 receptor 2* gene expression. There were positive correlations between *IP3 receptor 2* gene expression and IL-4 and IL-6 cytokine secretions. Our results conclude that CA might inhibit degranulation, IL-4 and IL-6 cytokine secretions, and *IP3 receptor 2* gene expression in C48/80-stimulated mouse P815 mast cells.

## 1. Introduction

Anaphylaxis, which may be induced by food and drug allergies, is a systemic and potentially life-threatening disease involving the skin, lungs, gastrointestinal tract, and cardiovascular system [1]. The triggering causes include IgE-mediated and non-IgE-mediated pathways, which may induce mast cell or basophil degranulation and subsequent allergic reaction mechanisms such as redness, swelling, itching, urticaria, angioedema, pain, nausea, vomiting, and respiratory complications [2]. In IgE-mediated anaphylaxis, specific IgE antibodies are first produced by B cells after antigens are recognized by T cells and induce a type 2 helper T cell (Th2) immune balance. The specific IgE will bind to high-affinity IgE receptors (FcεR I) on mast cells and basophils, promoting cell activation [3]. In non-IgE-dependent mechanisms to induce anaphylaxis, the activated binding, including IgG to FcγRII/III, complements to C3a/C5a receptors, and drugs to Mas-related G protein-coupled receptor-X2 (Mas-related G protein-coupled receptor, member X2, MRGP receptor X2), may further change mast cell growth, differentiation, migration, apoptosis, and degranulation [4].

Mast cells, when they are stimulated and activated, release the contents of the granules, including histamine, serotonin, proteoglycans, and proteases, which is called degranulation, and then synthesize and secrete lipid mediators, cytokines, and chemokines [5]. Many cationic substances have been found to initiate IgE-independent pathways, including compound 48/80 (C48/80), substance P (SP), and vasoactive intestinal peptide. Particularly, MRGP receptor X2, a receptor expressed in human mast cells for degranulation signaling, recently attracted much attention and is related to chronic diseases such as urticaria, atopic dermatitis, asthma, and rheumatoid arthritis [6,7]. *Mas-related G protein-coupled receptor, member B2* (*MRGP receptor B2*) in mice and *MRGP receptor B3* in rats, are orthologous genes of *MRGP receptor X2* [6,7]. Regardless of the IgE-dependent or IgE-independent pathway, the flow of calcium ions (Ca^2+^) is a vital response mechanism for the mast cell activation and degranulation, which may further involve many protein expression and phosphorylation reactions, including tyrosine-protein kinase Lyn, phospholipase Cγ (PLCγ), inositol-1,4,5-trisphosphate receptor (IP3 receptor), etc. [8,9,10]. Natural phytochemicals, such as caffeic acid phenethyl ester (CAPE), which is a caffeic acid (CA) derivative, have been reported to alleviate allergic inflammation through interfering with mast cell activation and degranulation signaling [11,12,13].

In our preliminary study, CA, 3-*O*-caffeoylquinic acid (3-CQA), and 5-*O*-caffeoylquinic acid (5-CQA) were suggested to have antiallergic potential among selected CA derivatives. However, the antiallergic actions and mechanisms of CA, 3-CQA, and 5-CQA remain unclear. Thus, antiallergic effects of CA, 3-CQA, and 5-CQA were explored to determine whether they inhibit the mast cell activation on mouse P815 mast cell degranulation by globally assessing the release of β-hexosaminidase, the appearance and morphological changes, and cytokine secretions. The most antiallergic potential compound to the stimulated P815 mast cells was further selected to evaluate the impact on the expression of degranulation-related pathway genes, including *MRGP receptor B2* and *inositol 1,4,5-triphosphate receptor 2* (*IP3 receptor 2*), using reverse transcription real-time quantitative polymerase chain reaction (qPCR).

## 2. Results and Discussion

### 2.1. Non-Cytotoxic Concentrations of CA, 3-CQA, and 5-CQA to the Mouse P815 Mast Cells

This study used an MTT assay to evaluate survival rates of mouse P815 mast cells to achieve non-cytotoxic doses of CA, 3-CQA, and 5-CQA for exploring their effects on degranulation. Figure 1 shows the treatment impact of CA, 3-CQA, and 5-CQA on the growth of mouse P815 mast cells. The results showed no statistically significant difference in the survival rate between CA treatments and the control when the treatment concentration was lower than 25 μM (Figure 1A). The 3-CQA treatments at a concentration of 25 μM significantly reduced (*p* < 0.05) the survival rate of the mast cells (Figure 1B), reflecting that the survival rate decreased to 71.1% compared with the control. The 5-CQA treatments at a concentration of 25 μM significantly reduced (*p* < 0.05) the survival rate of the mouse P815 mast cells (Figure 1C), reflecting that the survival rate decreased to 73.4% compared with the control. Compared with the survival rates, the results evidenced that CA had lower cytotoxicity to mast cells than 3-CQA and 5-CQA. Our results suggest that the optimal non-cytotoxic concentrations for the following mast cell degranulation experiments were 1, 5, 10, and 25 μM for CA, 1, 5, and 10 μM for 3-CQA, as well as 1, 5, and 10 μM for 5-CQA, respectively.

### 2.2. Degranulation Effects of C48/80, Ovalbumin (OVA), and 5-HMF on β-Hexosaminidase Release from Mouse P815 Mast Cells

To investigate whether food allergens such as OVA and 5-HMF can induce degranulation of mouse P815 mast cell line, and to understand whether caffeic acid and its derivatives have the potential to reduce the occurrence of food allergies, food allergens OVA (0.1, 0.5, and 1 mg/mL) and 5-HMF (0.5, 2.5, and 5 mM) were subjected to the mast cell degranulation experiment. Cell culture medium was selected as a negative control, and C48/80 as a positive control. The percentage of β-hexosaminidase released from the cells was adopted to evaluate whether food allergens have the potential to act as mast cell degranulation promoters. Figure 2 shows the effect of adding food allergens (OVA or 5-HMF) on the release of β-hexosaminidase. The results showed that OVA at a concentration of 1 mg/mL and 5-HMF at concentrations of 0.5–5 mM significantly (*p* < 0.05) increased the release of β-hexosaminidase from mast cells compared with the control group. The addition of OVA and 5-HMF increased the percentage of β-hexosaminidase release; however, both were significantly lower than C48/80. Moreover, 5-HMF had a significantly higher degranulation effect than OVA. OVA just had a weak activity for stimulating mast cells, but 5-HMF had more potential to induce mast cell degranulation and caused allergic phenomena. Therefore, 5 mM of 5-HMF and 90 μg/mL of C48/80 were selected as mast cell degranulation stimulators for subsequent experiments.

### 2.3. Inhibitory Effects of CA, 3-CQA, and 5-CQA Treatments on β-Hexosaminidase Release from Mouse P815 Mast Cells Stimulated with C48/80 or 5-HMF

Figure 3 shows the impact of CA, 3-CQA, and 5-CQA treatments on the release of β-hexosaminidase in the mast cells stimulated with C48/80. The results showed that C48/80 significantly increased (*p* < 0.05) β-hexosaminidase release from the stimulated mast cells compared to the vehicle control. Significantly, CA treatments reduced considerably (*p* < 0.05) the C48/80-induced mast cell β-hexosaminidase release compared to the control, suggesting the antiallergic potential of CA in stabilizing mast cell degranulation (Figure 3A). Among 3-CQA and 5-CQA treatments, 3-CQA and 5-CQA at 1 μM could significantly reduce (*p* < 0.05) β-hexosaminidase release compared to the controls (Figure 3B,C); however, there was no statistically significant difference at the higher concentrations (5 μM and 10 μM) compared with the controls. We supposed that the higher concentrations of 3-CQA and 5-CQA might have a mildly stimulatory effect on mast cells, slightly offsetting their antiallergic potential. Therefore, CA was selected to perform further in the subsequent experiment.

Figure 4 shows the effects of CA, 3-CQA, and 5-CQA treatments on the release of β-hexosaminidase in the mast cells stimulated with 5-HMF. The results showed that the release of β-hexosaminidase in the control group tended to increase compared with the vehicle control, but there was no statistically significant difference. A C48/80 treatment alone was selected for comparison in the experiment. The stimulatory effect on the mast cell degranulation was significantly higher (*p* < 0.05) than that of 5-HMF. Unfortunately, adding CA, 3-CQA, and 5-CQA could not considerably affect (*p* > 0.05) the percentage of β-hexosaminidase release in this 5-HMF-sensitized experiment model.

### 2.4. Effects of CA Treatments on the Appearance and Morphology of Mouse P815 Mast Cells Stimulated with C48/80 or 5-HMF

To understand the effect of CA on the appearance and morphology change in mouse P815 mast cells stimulated with C48/80 or 5-HMF, mast cells in each treatment were stained with TB dye, and the stained cell images were recorded using microscope-specific image acquisition and analysis software. The cell numbers observed in the view field and the degree of degranulation were quantified using image analysis software. Figure 5 shows the effect of CA treatments on the appearance and morphology of the mouse P815 mast cells stimulated with C48/80. The results showed that CA treatment in the 1–25 μM concentration range significantly increased (*p* < 0.05) the cell number observed in the view field (Figure 5G), indicating that CA may protect mast cells from C48/80 damage. Moreover, the degree of mast cell degranulation was evaluated based on the color change in mast cells. Compared with the control, the results showed that CA could significantly inhibit (*p* < 0.05) the degranulation of mast cells induced by C48/80 (Figure 5H).

Figure 6 shows the effect of CA treatments on the appearance and morphology of the mouse P815 mast cells stimulated with 5-HMF. CA treatment at 10 μM significantly (*p* < 0.05) increased the number of cells observed in the view field (Figure 6G), indicating that CA may protect mast cells from 5-HMF damage. Compared with the control, the results showed that CA treatment could not significantly inhibit (*p* > 0.05) the degranulation of mast cells induced by 5-HMF (Figure 6H).

Based on the experiments, CA has been evidenced to have the most potential to hinder mast cell degranulation; therefore, it was further subjected to analysis of the influence on Th2 (IL-4) and pro-inflammatory (IL-6) cytokines secreted by the mast cells stimulated with C48/80 or 5-HMF.

### 2.5. Effects of CA Treatments on Th2 (IL-4) and Pro-Inflammatory (IL-6) Cytokines Secretion by Mouse P815 Mast Cells Stimulated with C48/80 or 5-HMF

Table 1 shows the effects of CA treatments on cytokine secretions by the mouse P815 mast cells stimulated by C48/80 or 5-HMF. The results showed that C48/80 significantly increased (*p* < 0.05) IL-4 secretion and slightly increased (*p* > 0.05) IL-6 secretion compared to vehicle controls. CA treatments in the 5–25 μM concentration range significantly (*p* < 0.05) and dose-dependently reduced IL-4 and IL-6 secretions by C48/80-stimulated mast cells, exhibiting their antiallergic and anti-inflammatory potential.

Interestingly, the results showed that 5-HMF could not significantly (*p* > 0.05) influence the IL-4 secretion but significantly decreased (*p* < 0.05) the IL-6 secretion compared to vehicle controls (Table 1), suggesting that 5-HMF treatment alone has an anti-inflammatory potential for mast cells but is poor in sensitizing the mast cells. However, CA treatments slightly (*p* > 0.05) decreased IL-4 secretion but significantly (*p* < 0.05) and dose-dependently reduced IL-6 secretion compared to those of controls. Our results suggest that CA treatment enhances the 5-HMF’s anti-inflammatory effects compared to the control. The 5-HMF’s anti-inflammatory effect on mast cells remains to be further studied.

Compared with C48/80 and 5-HMF effects, C48/80 has a superior impact in causing allergic and pro-inflammatory effects on mast cells. Therefore, C48/80 was used to investigate degranulation-related signaling pathways.

### 2.6. Effects of CA Treatments on the Relative Expression Folds of MRGP Receptor B2 and IP3 Receptor 2 Genes in C48/80-Stimulated Mouse P815 Mast Cells

This study used real-time qPCR to analyze the changes in relative expression folds of degranulation-related signaling genes: *MRGP receptor B2* and *IP3 receptor 2* in C48/80-stimulated mouse P815 mast cells. Table 2 shows the effect of CA treatment on the relative expression folds of *MRGP receptor B2* and *IP3 receptor 2* genes in C48/80-stimulated mouse P815 mast cells. The results showed that the control group (C48/80 treatment alone) significantly increased *MRGP receptor B2* and *IP3 receptor 2* gene relative expression folds compared to vehicle controls. Notably, CA treatments significantly decreased (*p* < 0.05) *IP3 receptor 2* gene relative expression folds; however, they slightly (*p* > 0.05) but dose-dependently increased *MRGP receptor B2* gene relative expression folds compared with controls. Our results evidenced that CA treatment at 25 μM could inhibit *IP3 receptor 2* gene relative expression folds in C48/80-stimulated mast cells, thereby alleviating degranulation in the mast cells. CA may potentially be developed as an inhibitory agent for mast cell degranulation.

### 2.7. Associations Among Degranulation Degrees, Relative Gene Expression Folds, and Cytokine Scretions by Mast Cells

Pearson product–moment correlation coefficient (r) was adopted to analyze associations among degranulation degrees, relative expression folds of degranulation-related signaling pathway genes, and cytokine secretions by mast cells (Figure 3 and Figure 4). When the r value is positive, it represents a positive correlation. Otherwise, it represents a negative correlation. If the absolute r value is more significant than 0.7, it means a high correlation. When the absolute r value is between 0.3 and 0.7, it means a moderate correlation. If the absolute r value is below 0.3, it means a low correlation. Figure 7A shows the correlation between the relative expression folds of *MRGP receptor B2* and β-hexosaminidase release in P815 mast cells. The results showed a significantly (*p* < 0.05) positive correlation between the MRGP receptor B2 mRNA expression amount and the β-hexosaminidase release rate (r = 0.456, * *p* = 0.025). More obviously, there was a significant (*p* < 0.05) positive correlation between the IP3 receptor 2 mRNA expression amount and the β-hexosaminidase release rate (r = 0.523, ** *p* = 0.009) (Figure 7B). Compared with the MRGP receptor B2 mRNA expression amount, a higher positive correlation between the IP3 receptor 2 mRNA expression amount and the β-hexosaminidase release rate was found (Figure 7). Importantly, CA treatment had a significantly inhibitory effect on the relative expression folds of the *IP3 receptor 2* gene (Table 2). Therefore, *IP3 receptor 2* gene relative expression folds were further selected to analyze the correlation with the cytokines (IL-4 and IL-6) levels secreted by the mast cells (Figure 8). Figure 8 shows correlations between the relative expression folds of the *IP3 receptor 2* gene and the cytokines (IL-4 and IL-6) secreted by the mast cells. The results showed that the relative expression folds of the *IP3 receptor 2* gene are significantly (*p* < 0.05) correlated with IL-4 (r = 0.612, ** *p* = 0.000) and IL-6 (r = 0.383, * *p* = 0.03) cytokine secretions (Figure 8). Based on the results, it is speculated that CA treatment may reduce the relative expression folds of the *IP3 receptor 2* gene (Table 1), consequently inhibiting the release of β-hexosaminidase (Figure 3A) and the secretion of IL-4 and IL-6 cytokines (Table 1), thereby slowing down the degranulation of C48/80-stimulated mast cells. However, the weak link is between decreased IL-4 and IL-6 cytokine secretions and decreased IP3 receptor 2 gene expression in compound 48/80-stimulated mouse P815 mast cells. More experiments are needed to provide solid evidence in the future.

Mast cells are closely related to acute and chronic allergic diseases, and the cytokines they secrete play a key role in the symptoms of allergy and inflammation. Mast cells can synthesize and secrete various cytokines, such as IL-4, IL-5, and IL-6. For activated mast cells, IL-4 is one of the representative cytokines released by them, and it promotes the production of IL-13 and affects the immune balance toward the Th2 immune response [14]. During degranulation, mast cells also secrete IL-6, which is considered a pro-inflammatory cytokine [15]. The literature has shown that cytokines IL-4 and IL-6, which are highly associated with the Th2 and pro-inflammatory responses, promote the extent of allergic inflammation in cells [16]. Using a stimulated mast cell degranulation model, this study evaluated the antiallergic effects of CA, 3-CQA, and 5-CQA. CA’s inhibitory effect on β-hexosaminidase release was better than that of 3-CQA and 5-CQA (Figure 4 and Figure 6), indicating that CA has the most significant potential in soothing mast cells. CA or its caffeoylquinic acids (CQAs) have also been found to have the potential to protect against cardiovascular disease, diabetes, liver disease, neurodegenerative disease, intestinal inflammation, oxidative stress-related pathologies, and allergic diseases [17,18,19,20,21,22,23,24]. This study adopted different doses of CA to explore its effects on cytokine secretion by the stimulated mouse P815 mast cells. We evidenced that CA could significantly inhibit the secretion of IL-4 and IL-6 in a dose–response manner (Table 1), further illustrating that CA can potentially increase the stability of stimulated mast cells in the later stage by regulating various pro-inflammatory indicators, such as cytokines [25]. We hypothesized that CA’s lower molecular weight might help it to enter the mast cells to exert its immunomodulatory effects by decreasing the expression of *IP3 receptor 2* (Table 2), consequently inhibiting β-hexosaminidase release (Figure 3A) and the secretion of IL-4 and IL-6 cytokines (Table 1), thereby slowing down the degranulation of mast cells stimulated by C48/80 (Figure 9).

One commonly and traditionally used mast cell degranulation promoter is C48/80 [19]. In this experiment, we have evidenced the stimulatory effect of C48/80 on the mouse P815 mast cells, resulting in almost 50% β-hexosaminidase release (Figure 2) and convincing this suitable experimental model for exploring degranulation-related signaling pathways. C48/80 activates phospholipase D through heterotrimeric GTP-binding protein, increases the intracellular Ca^2+^ concentration, and then induces mast cell degranulation after C48/80 contacts the mast cell membrane [26]. Mast cell activation induced by C48/80 is closely related to the flow of Ca^2+^ and involves many related signal transductions, including the hydrolysis of phosphatidylinositol-4,5-bisphosphate (PIP2) by PLCγ to form IP3 and diacylglycerol (DAG). IP3 receptors are the receptors of IP3 in the endoplasmic reticulum (ER) membrane to mediate the Ca^2+^ release from intracellular stores in the ER in response to extracellular stimulants in almost all cells [27,28,29,30,31,32]. When IP3 and its receptor combine, the ER releases Ca^2+^. After the Ca^2+^ stored in the cell is consumed, it may promote Ca^2+^ to flow into the cell through transient receptor potential canonical (TRPC) or store-operated calcium entry channel (ORAI). Studies have shown that inhibiting IP3 receptor protein expression is related to reducing Ca^2+^ influx, which can hinder Ca^2+^ influx and protect activated mast cells by decreasing the expression of Ca^2+^-related channel proteins, such as TRPC1, ORAI1, stromal interaction molecule 1 (STIM1), PLCγ, and IP3 receptors [33,34]. In addition, the literature indicates that mast cell degranulation is related to the ATP content produced by mitochondria [9]. Chrysin has been found to inhibit the phosphorylation of IP3 and PLC with a dose-dependent effect. The binding of IP3 to the IP3 receptor will promote the flow of Ca^2+^, and the flow of Ca^2+^ in the ER also involves the activation of mitochondrial-related pathways [9]. Importantly, this study discovered that CA treatment in advance could significantly (*p* < 0.05) reduce the *IP3 receptor 2* gene expression in the C48/80 80-stimulated mast cells (Table 2). A significant (*p* < 0.05) positive correlation is observed between *IP3 receptor 2* gene expression and β-hexosaminidase release in the mast cells (Figure 7B). Moreover, a significant (*p* < 0.05) positive correlation exists between *IP3 receptor 2* gene expression and IL-4, as well as IL-6 cytokine secretions in the mast cells (Figure 8). However, CA treatment significantly inhibited the secretion of IL-4 and IL-6 cytokines by the mouse P815 mast cells stimulated by C48/80 (Table 1). It is speculated that CA treatment may inhibit the β-hexosaminidase release and IL-4 as well as IL-6 cytokines secretions by slowing down the gene expression of *IP3 receptor 2*, which has a high affinity for IP3 and serves as a Ca^2+^ release channel to reduce the amount of Ca^2+^ released in the ER, thereby alleviating the degranulation of mast cells stimulated by C48/80 (Figure 9). The inhibitory mechanism via molecular docking analysis was supposed to be that CA binds to IP3 receptor 2 in the C48/80 binding site region, preventing mast cell degranulation and alleviating mast cell reactions [35]. However, it should be further explored. The data from this experiment imply that CA can be an inhibitory agent of mast cells for treating allergic and inflammatory diseases.

CA has been found to possess diverse positive biological potential, particularly in neuroprotective and anti-inflammatory effects, anti-diabetic, anti-atherosclerotic, anti-cancer, and anti-Alzheimer’s diseases [36,37,38,39,40]. In terms of antiallergic effects, CA has been relatively less studied. However, CA has been found to inhibit compound 48/80 and OVA-induced allergic symptoms in allergic dermatitis and allergic asthma in vivo [19,20]. The present study further unravels the antiallergic mechanisms that CA treatment may alleviate β-hexosaminidase release, IL-4 and IL-6 cytokine secretions, and *IP3 receptor 2* gene expression, thereby reducing the degrees of degranulation and inflammation status in C48/80-stimulated mouse P815 mast cells.

The present study has achieved some valuable data for evidencing CA’s antiallergic potential; however, there are a few limitations that remain to be further investigated. First, the experiment is merely an in vitro bioassay exploration; thus, more in vivo studies should be performed to harvest sound evidence for CA clinical uses. Potential confounding factors in this in vitro experiment, such as the CA’s cytotoxicity at higher concentrations, might activate mast cells and decrease therapeutic effects. Therefore, CA treatments in vivo on allergic diseases should be investigated by established animal studies [41,42,43]. To better assess the clinical relevance of CA as a mast cell inhibitor, in vivo studies (e.g., animal models of allergic inflammation) should be included to validate the therapeutic potential of CA. More molecular regulatory mechanisms of CA in activated mast cells, including TNF-α secretion, nuclear factor-κB (NF-κB) and cyclooxygenase-2 (COX-2) expression, and activation of mitogen-activated protein kinases (MAPKs) signaling pathway, should be involved [44]. Second, the degranulation mediators contain various components; in addition to β-hexosaminidase and cytokines, histamine or leukotrienes can also be measured to enhance the evidence of CA’s anti-mast cell degranulation. Third, more degranulation-related signaling pathways, except the MRGP receptor B2 and IP3 receptor 2 pathways, should be further explored to unravel the physiological and pharmacological functions of CA against mast cell degranulation globally. We proved that CA might independently inhibit degranulation, IL-4, and IL-6 cytokine secretions, as well as *IP3 receptor 2* gene expression in compound 48/80-stimulated mouse P815 mast cells; however, the inference of decreased IL-4 and IL-6 cytokine secretions with decreased *IP3 receptor 2* gene expression in compound 48/80-stimulated mouse P815 mast cells was still weak. It needs more molecular regulation in *IP3 receptor 2* gene-knockout (KO) cells and *MRGP receptor B2*-KO cells [45,46,47]. Additionally, it is necessary to detect the protein expression and active form levels of the gene in the future. Recovering the weaknesses to obtain more solid scientific data is helpful for a global and comprehensive understanding of the CA characteristics in further anti-allergenic applications.

## 3. Materials and Methods

### 3.1. Preparation of CA, 3-CQA, and 5-CQA Preparation

Caffeic acid (CA, C_9_H_8_O_4_, Sigma, C0625, St. Louis, MO, USA), 3-*O*-caffeoylquinic acid (3-CQA, C_16_H_18_O_9_, Cayman Chemical, NSC 407296, Ann Arbor, MI, USA), and 5-*O*-caffeoylquinic acid (5-CQA, C_16_H_18_O_9_, ChemFaces, CFN97472, Wuhan, China) were bought at the highest available purity (>98.0%) for the experiment [48,49]. CA, 3-CQA and 5-CQA were dissolved in dimethyl sulfoxide (DMSO, Tedia, DS1461-001, Fairfield, CT, USA) to prepare stock solutions at the indicated concentrations of 5 mM, 4 mM, and 4 mM, respectively. The stock solutions were sterilized with a 0.20 μm filter (Sartorius, Minisart NY, 17845-ACK, Goettingen, Germany) and diluted with a large amount of culture medium (CM) or Tyrode’s buffer to reduce DMSO (<1%) residue in the medium when used to eliminate the possible cytotoxicity of DMSO [48,49]. The preliminary studies have reported that DMSO concentration lower than 1% in the medium did not induce any apparent cytotoxicity [48,49,50]. Since the maximal remaining DMSO concentrations were lower than 1% in the medium in the experiment, a DMSO control group was not included in the cytotoxic experiment.

### 3.2. Culture of the Mouse P815 Mast Cell Line

#### 3.2.1. Source of Mouse P815 Mast Cell Line

The experiment was carried out with the P815 mast cell line. The major information of Research Resource Identifier (RRID) concerning the mast cell line is provided as follows: https://catalog.bcrc.firdi.org.tw/BcrcContent?bid=60178&rowid=1 (accessed on 8 September 2021). P815 cell line is a mast cell line that was originally isolated from a tumor with a disease of mastocytoma (mast cell). P815 cell is an animal cell from *Mus musculus* (DBA/2 mouse) mast cell; mastocytoma; mouse mastocytoma [51]. It can be applied in immunological research with 3D cell culture. The cell type of the P815 cell line is a mast cell that can be classified as Eukaryota, Animalia, Metazoa, Chordata, Vertebrata, and Tetrapoda. The mast cell line was purchased from the Bioresource Collection and Research Center (BCRC, BCRC No. 60178) of the Food Industry Research and Development Institute (FIRDI), Hsinchu, Taiwan, https://catalog.bcrc.firdi.org.tw/BcrcContent?bid=60178&rowid=1 (accessed on 8 September 2021). The P815 mast cell line, a specific type of immune cell and mast cell, was used in the experiment to explore allergic reactions, degranulation effects, inflammation, and immunological research [52]. The P815 mast cell line used in the experiment was confirmed to be free of contamination by Mycoplasma.

#### 3.2.2. Cell Thawing, Subculturing, and Cryopreservation

The cryovial containing P815 mast cells was removed from liquid nitrogen and placed in a 37 °C water bath for rapid thawing. After the cells were thawed entirely, they were pipetted out and put in a 10 cm cell culture dish (Thermo Scientific, 150466, Suzhou, China) containing 10 mL of CM consisting of 90% DMEM/High Glucose which contains 4 mM L-glutamine, 4.5 g/L glucose and 1.5 g/L sodium bicarbonate (HyClone SH30022.02, South Logan, UT, USA), 10% of Fetal Bovine Serum (FBS, Gibco 10437-028, Grand Island, New York, NY, USA), and 1% of penicillin–streptomycin antibiotic solution (Gibco 15140-122, Grand Island, New York, NY, USA) [53]. After gently shaking, the cells were cultured in a 37 °C and 5% CO_2_ incubator (Shel lab TC2323, Cornelius, OR, USA) for 24 h. The medium was replaced with a fresh CM to remove the original DMSO culture medium. The cells were continuously cultured and subcultured at an appropriate subdivision ratio when they reached 90% confluence. The cell generations used in the experiments were maintained at passage numbers between 4 and 10.

The mouse P815 cell line is characterized as a suspension cell, but some cells are adherent. Different types of mast cells need to be collected in two different ways during subculturing. The suspended cells were first harvested by pipetting the medium in the Petri dish into a sterile 15 mL centrifuge tube. Aliquots of 10 mL of phosphate-buffered saline (PBS, 8.1 mmoL/L Na_2_HPO_4_, 1.5 mmoL/L KH_2_PO_4_, 137 mmoL/L NaCl, 2.7 mmoL/L KCl, pH 7.4, autoclaved) were added to the Petri dish to rinse the cell surface twice to remove the residual medium and cellular metabolic waste. An aliquot of 0.5 mL of 0.25% trypsin (10×, Corning, 25-054, New York, NY, USA) in PBS was added to the Petri dish, incubated at 37 °C and 5% CO_2_ for another 4 min, and then observed under a microscope (Olympus, CKX31, Tokyo, Japan) to confirm that the cells had fallen off from the bottom of the Petri dish. An aliquot of 1.5 mL CM was added to neutralize trypsin. The cell suspension was pipetted and collected in a sterile 15 mL centrifuge tube. The tube was centrifuged at 25 °C and 400× *g* for 5 min. The supernatant was carefully removed. The cell pellet was harvested, added with 1 mL of CM, and thoroughly mixed. The cell density was counted using Trypan Blue (Coring, 25-900-C1, Manassas, VA, USA) exclusion assay with a hemocytometer, and the cell density was adjusted to 10^5^–10^6^ cells/mL. The cells were used for mast cell degranulation experiments in the present study. In our preliminary study, the doubling time of the P815 mast cell line was estimated at 23.5 ± 3.1 h, suggesting that the most suitable culture time falls in 24–48 h.

To permanently freeze-store the cells, the mast cells at the log growth phase were harvested, adjusted for cell density, added to the cryopreservation solution (93% CM, 7% DMSO), and mixed evenly in a cryovial. Finally, each cryovial contained 1 mL (2 × 10^6^ cells/mL) of mouse P815 mast cells and was appropriately labeled. The cells were progressively frozen in a 4 °C refrigerator for 20 min, a −30 °C refrigerator for 20 min, and a −80 °C refrigerator for 24 h, and then moved to a liquid nitrogen tank for long-term storage.

### 3.3. Effects of Treatments with CA, 3-CQA, and 5-CQA on the Viability of Mouse P815 Mast Cells

To assess the treatment effects of CA, 3-CQA and 5-CQA on the viability of P815 mast cells for determining optimal non-cytotoxic concentrations, P815 mast cells (1 × 10^5^ cells/mL, 100 μL/well) were transferred into each well of the 96-well plate and incubated at 37 °C and 5% CO_2_ for 24 h. The 96-well plate was centrifuged for 10 min at 25 °C and 400× *g*, and the supernatant was carefully removed. Aliquots of 100 μL of test samples at different concentrations were added to the cells and cultured for 24 h. The P815 mast cell viability was assessed using a 3-(4,5-dimethylthiazol-2-diphenyl)-2,5-tetrazolium bromide (MTT) assay. Briefly, each well in the 96-well plate was filled with an aliquot of 10 μL MTT (5 mg/mL PBS, Sigma, M-5655, St. Louis, MO, USA). The plate was incubated for an additional 4 h at 37 °C and 5% CO_2_. The plate was centrifuged for 10 min at 25 °C, 400× g. The supernatant in each well was carefully removed, and the cell pellet was washed twice with 100 μL/well PBS buffer. An aliquot of 100 μL/well DMSO was pipetted to each well and gently shaken for 30 min. The absorbance at a wavelength of 550 nm was measured using a microplate ELISA reader (FLUOstar Omega, 415-1103, BMG Labtech, Offenburg, Germany). The absorbance accords with viable cell numbers and can measure the cell viability influenced by the test sample. The viability of P815 mast cells was displayed as the percentage of survival rate in contrast to the mean absorbency (A) of the control. The cell viability was measured by the equation: survival rate (% of control) = [(A_sample_ − A_blank_)/(A_control_ − A_blank_)] × 100 [54,55].

### 3.4. Effects of C48/80, Ovalbumin (OVA), and 5-HMF on β-Hexosaminidase Release from Mouse P815 Mast Cells

C48/80 (Sigma, C2313, St. Louis, MO, USA) was evidenced to be a potent degranulation agent. Ovalbumin (OVA, Sigma, A5378, St. Louis, MO, USA), which is recognized as an allergic sensitizer, might have stimulatory potential to induce mast cell degranulation. 5-Hydroxymethyl-2-furaldehyde (5-HMF, Sigma, H40807, St. Louis, MO, USA), derived from the Maillard reaction, caramelization reaction, heat processing, or long-term storage, may cause mast cell degranulation. To compare the possible degranulation effect, C48/80, OVA, and 5-HMF were selected as stimulatory agents for comparison in promoting mast cell degranulation in this experiment. P815 mast cells (1 × 10^5^ cells/mL, 500 μL/well) were transferred to the well of the 24-well plate and incubated for 24 h at 37 °C and 5% CO_2_. After incubation, the plate was centrifuged for 10 min at 25 °C and 400× *g*. The supernatant was carefully removed, and aliquots of 500 μL/well of Tyrode’s buffer were added to each well to wash once. Tyrode’s buffer (pH 7.4, 0.22 μm filter filtered) was composed of 10 mM HEPES (Biological Industries, 03-025-1B, Kibbutz Beit Haemek, Israel), 135 mM NaCl (Wako, 191-01665, Osaka, Japan), 5 mM KCl (Wako, 163-03545, Osaka, Japan), 1.8 mM CaCl_2_, 1 mM MgCl_2_ (Sigma, M8266, St. Louis, MO, USA), 5.6 mM Glucose (MP Biomedical, Inc., 152527, Eschwege, Germany), and 0.1% Bovine serum albumin (BSA, Thermo, Prod23209, Rockford, IL, USA) in 1000 mL deionized water. Then, aliquots of 200 μL/well of Tyrode’s buffer, C48/80 (90 μg/mL), OVA (0.1, 0.5, and 1.0 mg/mL), and 5-HMF (0.5, 2.5, and 5 nM) were, respectively, added to the cells and incubated for 30 min at 37 °C and 5% CO_2_. The plate was incubated on ice for 10 min to terminate the reaction. The plate was centrifuged for 10 min at 25 °C and 400× *g*. The supernatant was harvested and collected into a clean microcentrifuge tube. The cell pellet in each well was added with 200 μL/well 0.1% Triton X-100 solution (Sigma, 93443, St. Louis, MO, USA) and shaken gently for 30 min to lyse the cells at room temperature. Then, the plate was centrifuged for 10 min at 25 °C and 400× *g*. The cell lysate in each well was harvested and collected into another clean microcentrifuge tube. Aliquots of 50 μL of the supernatant and cell lysate were pipetted into the well in the 96-well plate. Then, 50 μL/well 2 mM *p*-nitrophenyl-N-acetyl-β-D-glucosaminide (P-NAG, Sigma, N9376, St. Louis, MO, USA) was added to each well and incubated for 1 h at 37 °C and 5% CO_2_. Finally, aliquots of 200 μL/well 0.1 M Na_2_CO_3_/NaHCO_3_ buffer (pH 10.0) were added to each well to terminate the reaction. The optical density (OD) at 405 nm was measured, and the effects of the test sample on the release of β-hexosaminidase were calculated with the formula [41,56]:β-hexosaminidase release (%) = [OD_405_ supernatant/(OD_405_ supernatant + OD_405_ cell lysate)] × 100

### 3.5. Effects of Treatments with CA, 3-CQA, and 5-CQA on β-Hexosaminidase Release from Mouse P815 Mast Cells Stimulated by C48/80 or 5-HMF

P815 mast cells (1 × 10^5^ cells/mL, 500 μL/well) were transferred to the well of the 24-well plate and incubated at 37 °C and 5% CO_2_ for 24 h. The plate was centrifuged for 10 min at 25 °C and 400× *g*, the supernatant was carefully removed, and aliquots of 500 μL/well of Tyrode’s buffer were added to each well to wash once. Then, aliquots of 160 μL/well of Tyrode’s buffer were added to the cells and incubated for 10 min at 37 °C and 5% CO_2_. Aliquots of 20 μL/well test samples (CA, 3-CQA, and 5-CQA) at different concentrations were added to the cells and cultured for 30 min at 37 °C and 5% CO_2_. Aliquots of 20 μL/well C48/80 (final concentration was 90 μg/mL in the well) or 5-HMF (final concentration in the well: 5 mM) were added to the cells. The cells were incubated at 37 °C and 5% CO_2_ for 30 min, and they were then incubated on ice for 10 min to terminate the reaction. The plate was centrifuged for 10 min at 25 °C and 400× *g*. The supernatant was harvested and collected into a clean microcentrifuge tube. The cell pellet in each well was added with 200 μL/well 0.1% Triton X-100 solution and shaken gently for 30 min to lyse the cells at room temperature. Then, the plate was centrifuged for 10 min at 25 °C and 400× *g*. The cell lysate in each well was harvested and collected into another clean microcentrifuge tube. Aliquots of 50 μL of the supernatant and cell lysate were pipetted into the well in the 96-well plate. Then, 50 μL/well 2 mM *p*-nitrophenyl-N-acetyl-β-D-glucosaminide was added to each well and incubated for 1 h at 37 °C and 5% CO_2_. Finally, aliquots of 200 μL/well 0.1 M Na_2_CO_3_/NaHCO_3_ buffer (pH 10.0) were added to each well to terminate the reaction. The optical density (OD) at 405 nm was measured, and the effects of the test sample on the release of β-hexosaminidase were calculated with the formula [41,56]:β-hexosaminidase release (%) = [OD_405_ supernatant/(OD_405_ supernatant + OD_405_ cell lysate)] × 100Inhibition rate (%) = [(control _β-hexosaminidase release (%)_ − sample _β-hexosaminidase release (%)_)/(control _β-hexosaminidase release (%)_ − vehicle control _β-hexosaminidase release (%)_)] × 100

### 3.6. Effects of CA Treatments on the Appearance and Morphology Change in the Mouse P815 Mast Cell Line Stimulated with C48/80 or 5-HMF

Toluidine Blue (TB) is an acidophilic metachromatic dye that selectively stains acidic components and has a high affinity for nucleic acids. It attaches to DNA or RNA to appear blue and to mast cell granules to appear purple; therefore, TB is an ordinary stain for mast cells [57]. The experiment used TB to stain mouse P815 mast cells to observe the morphological changes. Briefly, P815 mast cells (1 × 10^5^ cells/mL, 1 mL/well) were transferred to the well of the 12-well plate and incubated for 24 h at 37 °C and 5% CO_2_. After incubation, the plate was centrifuged for 10 min at 25 °C and 400× *g*. The supernatant was carefully removed. The cell pellet in the well was added with 360 μL/well CM, and the test sample (CA) in CM at different concentrations. The plate was incubated for 30 min at 37 °C and 5% CO_2_. Then, 40 μL/well C48/80 (final concentration in the well: 90 μg/mL) or 5-HMF (final concentration in the well: 5 mM) was added to the cells and incubated for 30 min at 37 °C and 5% CO_2_. The plate was centrifuged at 25 °C and 800× *g* for 10 min, and the supernatant was removed. The cells were fixed with 95% ethanol at room temperature for 30 min. The remaining 95% ethanol was removed and air-dried. The cells were stained with 0.1% TB (Toluidine Blue O, Sigma, 198161, St. Louis, MO, USA) dye (pH 2.5) in 1% NaCl solution for 30 min. The remaining TB dye was removed, and the cells were rinsed with PBS twice. The appearance and morphology of the cells were observed using an inverted microscope, and the cell images were recorded using microscope-specific image capture and analysis software. The image analysis (ImageJ, version 1.53) software was then used to quantify the cell numbers and degranulation degrees observed in the microscope field. The literature indicates that mast cells have intact, smooth cell membranes without being stimulated and activated. These abundant and evenly distributed granular substances can be observed in the cells. However, mast cells that have been stimulated with C48/80 to degranulate induce changes in cell morphology, such as cell deformation and contraction, the release of granular substances, low granule content in the cytoplasm, and light color of cells after TB staining [41,56].

### 3.7. Effects of CA Treatments on Cytokine Secretions by Mouse P815 Mast Cells Stimulated with C48/80 or 5-HMF

P815 mast cells (1 × 10^5^ cells/mL, 1 mL/well) were transferred to the well of the 48-well plate and incubated for 24 h at 37 °C and 5% CO_2_. The plate was centrifuged for 10 min at 25 °C and 400× *g*. The supernatant was carefully removed. The cell pellet in the well was added with aliquots of 360 μL/well CM, and the test sample (CA) in CM at different concentrations. The plate was incubated for 1 h at 37 °C and 5% CO_2_. Then, 40 μL/well C48/80 (final concentration in the well: 90 μg/mL) and 5-HMF (final concentration in the well: 5 mM) were added to the cells and incubated for 6 h at 37 °C and 5% CO_2_. The plate was centrifuged for 10 min at 25 °C and 400× *g*. The supernatant was harvested and collected into a microcentrifuge tube for determining IL-4 and IL-6 cytokine secretion levels [58]. The secretion profiles of IL-4 (R&D Systems, DuoSet ELISA Development Kits DY404, Minneapolis, MN, USA) and IL-6 (R&D Systems, DuoSet ELISA Development Kits DY406, Minneapolis, MN, USA) cytokines were measured by enzyme-linked immunosorbent assay (ELISA) according to the manufacturer’s instructions [41].

### 3.8. Effects of CA Treatments on the Expression of Degranulation-Related Pathway Genes in Mouse P815 Mast Cells Stimulated with C48/80

#### 3.8.1. Cell Culture for Total RNA Extraction

P815 mast cells (1 × 10^5^ cells/mL, 2 mL/well) were transferred to the well of the 6-well plate and incubated for 24 h at 37 °C and 5% CO_2_. After incubation, the plate was centrifuged for 10 min at 25 °C and 450× *g*. The supernatant was carefully removed. The cell pellet in the well was washed once with 1 mL/well PBS. The cell pellet in the well was added with 720 μL/well CM, and the test sample (CA) in CM at different concentrations. The plate was incubated for 1.5 h at 37 °C and 5% CO_2_. Then, 80 μL/well C48/80 (final concentration in the well: 90 μg/mL) was added to the cells and incubated at 37 °C and 5% CO_2_ for 1.5 h. The plate was incubated on ice for 10 min to stop the reaction. The plate was centrifuged at 25 °C and 450× *g* for 10 min, and the supernatant was carefully discarded. Aliquots of 0.8 mL/well PBS were added to each well to wash once. The remaining cell pellet in each well was immediately subjected to extracting total RNA with Trizol reagent.

Briefly, 1 mL/well Trizol reagent (Ambion, 157707, Carlsbad, CA, USA) was pipetted into each well to lyse and homogenize the cell pellet. The lysed cell pellet was harvested and collected into a 1.5 mL sterile microcentrifuge tube to extract total RNA and perform as described previously [41,49].

#### 3.8.2. Synthesis of First-Strand Complementary DNA (cDNA) from the Extracted Total RNA

Briefly, an aliquot of 1 µg total RNA was pipetted into a clean microcentrifuge tube to synthesize the first-strand cDNA using RNase-free DNase kit (Thermo, EN0521, Waltham, MA, USA) and M-MLV Reverse Transcriptase kit (Invitrogen, 28025-013, Carlsbad, CA, USA) and performed as described previously [41,49]. The prepared cDNA samples were subjected to a real-time quantitative PCR (qPCR) assay.

#### 3.8.3. Determination of the Relative Expression Amount of Degranulation-Related Genes, MRGP Receptor B2, and IP3 Receptor 2 Using Real-Time qPCR

Briefly, the prepared cDNA samples were adjusted to the same concentration with nuclease-free water (Promega, P1193, Madison, WI, USA), respectively. The primers of target genes in mouse P815 mast cells were devised using online software according to the corresponding cDNA sequences, www.ncbi.nlm.nih.gov (accessed on 5 January 2022). The primers of the target gene-specific PCR were designed as follows: *MRGP receptor B2* (forward primer, FW, 5′→3′): CCCTGGTTGGGATGGGACTA, *MRGP receptor B2* (reverse primer, RV, 5′→3′): GAAAGTCAGCCATAGCCAGGT; *IP3 receptor 2* (FW, 5′→3′): CCTGACGGTGAACAAGAGGTT, *IP3 receptor 2* (RV, 5′→3′): CAGCATTCACATTATCACCCTCG; β-actin (FW, 5′→3′): GCTACAGCTTCACCACCACA, β-actin (RV, 5′→3′): AAGGAAGGCTGGAAAAGAGC. Then, a total of 20 μL mixture, including nuclease-free water, qPCR Kit Master Mix (KAPA SYBR FAST qPCR Kit Master Mix, Kapa Biosystems, West Cummings Park, Suite Woburn, MA, USA), target gene-specific PCR primers, and diluted cDNA (cDNA template), was mixed and spun down. The qPCR reactant mixtures were performed and quantitated using a real-time rotary analyzer (Corbett Research Rotor-Gene 6000, Corbett Life Science, Sydney, Australia) with the following program: hot-start activation at 95 °C for 15 min, followed by 45 cycles of denaturation at 95 °C for 3 s, annealing at 65 °C for 20 s, extension at 72 °C for 15 s. Changes in MRGP receptor B2 and IP3 receptor 2 mRNA expression levels in the stimulated P815 mast cells were measured depending on the detected threshold cycle number (Ct) [41,49]. Six biological replicates were performed in two separate experiments to collect the essential data. The lower the Ct value, the higher the expression amount of the target gene.

#### 3.8.4. Data Calculations and Expressions

The Ct value is defined as the threshold cycle number of the target gene expression detected by fluorescence intensity using the real-time qPCR analyzer. In the experiment, a housekeeping gene, mouse β-actin, was adopted as a reference gene for comparison. MRGP receptor B2 and IP3 receptor 2 mRNA expressed in P815 mast cells treated with the test sample were computed. Their relative expression ratios (R) of mRNA in the stimulated mast cells are displayed as fold change values. Treated cells’ target mRNA relative expression amounts were calculated using the equation R = 2^−ΔΔCt^, where ΔCt = (Ct, target gene − Ct, reference gene) for first adjusting each test sample amount; ΔΔCt = (ΔCt, treatment x − ΔCt, control) = (Ct, target gene − Ct, β-actin), treatment x − (Ct, target gene − Ct, β-actin), control, for assessing each target gene expression (e.g., *MRGP receptor B2* or *IP3 receptor 2*) concerning its control condition. Finally, the equation R = 2^−ΔΔCt^ was adopted to compute the relative target mRNA expression ratio (R) between treated and control cells [41,49,54,59].

### 3.9. Statistical Analyses

The experimental data are presented as means ± standard deviation (SD). One-way analysis of variance (ANOVA) was used to analyze the data. Then, Duncan’s new multiple range test was adopted to test group differences. *p* < 0.05 represents a significant difference among groups. Pearson product–moment correlation coefficient (r) was used to analyze correlations between relative expression folds of degranulation-related genes and different parameters in activated mast cells. IBM SPSS Statistics version 20.0 software was adopted for performing statistical analysis.

## 4. Conclusions

The present study evidenced that CA was more effective than 3-CQA and 5-CQA in inhibiting the β-hexosaminidase release rate. CA treatment could increase the intact mast cells and reduce the release of granular substances from stimulated mast cells. CA significantly inhibited the secretion of IL-4 and IL-6 cytokines by C48/80- or 5-HMF-stimulated mast cells. Among CA, 3-CQA, and 5-CQA, CA is the most beneficial in inhibiting mast cell degranulation, protecting cells, and increasing their stability. CA could reduce the amount of *IP3 receptor 2* gene expression in the C48/80-stimulated mast cells, and the amount of *IP3 receptor 2* gene expression positively correlates with the β-hexosaminidase release, IL-4, and IL-6 cytokine secretions, respectively. Our results concluded that CA treatment may inhibit β-hexosaminidase release, IL-4 and IL-6 cytokine secretions, and *IP3 receptor 2* gene expression, thereby reducing the degrees of degranulation and inflammation status in C48/80-stimulated mouse P815 mast cells.

## Figures and Tables

**Figure 1 ijms-26-10086-f001:**
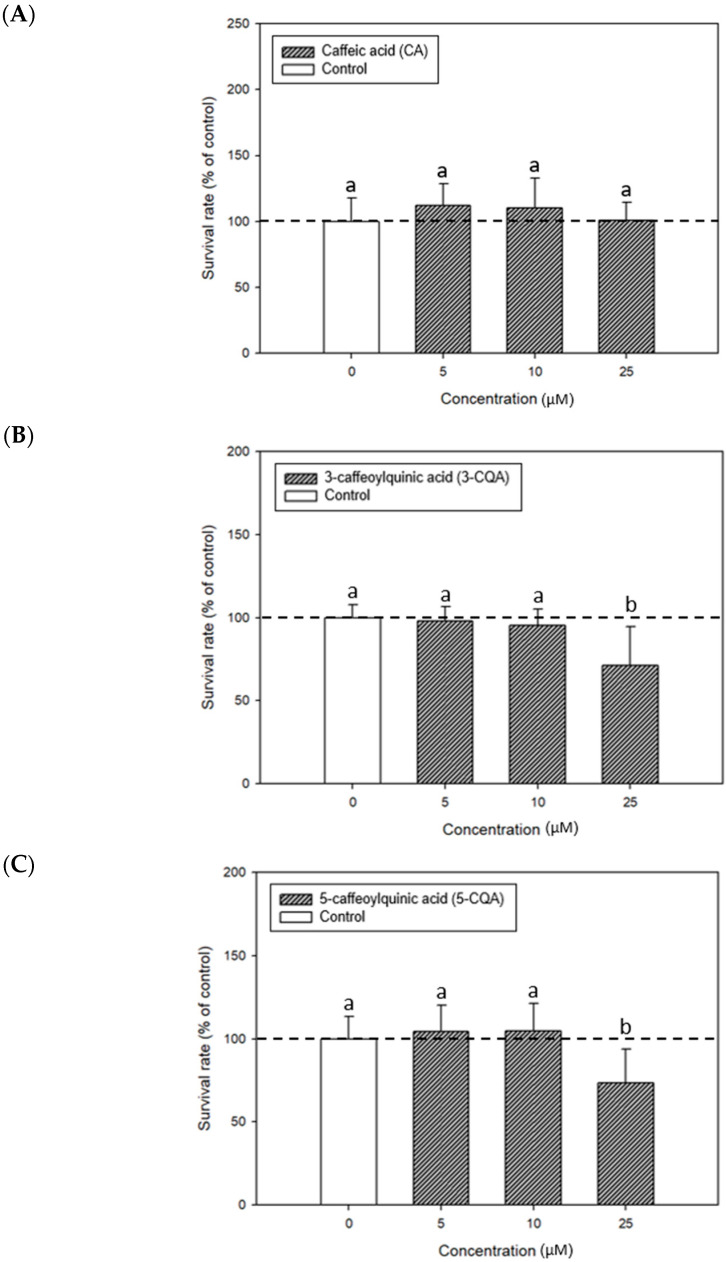
Effects of CA (**A**), 3-CQA (**B**), and 5-CQA (**C**) on the mouse P815 mast cell growth. Values are means ± SD (*n* = 6 biological determinations). Bars in the same plot not sharing a common letter (a, b) are significantly different (*p* < 0.05) from each other, analyzed by one-way ANOVA, followed by Duncan’s multiple range test. The original cell density was 1 × 10^5^ cells/mL.

**Figure 2 ijms-26-10086-f002:**
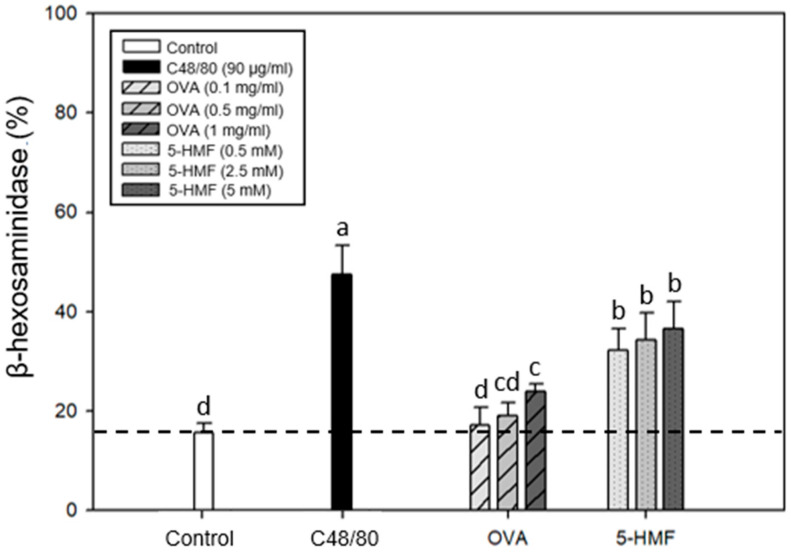
Effects of C48/80, OVA, and 5-HMF treatments on the β-hexosaminidase release of mouse P815 mast cells. Values are means ± SD (*n* = 6 biological determinations). Bars not sharing a common letter are significantly different (*p* < 0.05) from each other, analyzed by one-way ANOVA, followed by Duncan’s multiple range test. The original cell density was 1 × 10^5^ cells/mL. C48/80, compound 48/80; OVA, ovalbumin; 5-HMF, 5-hydroxymethyl-2-furaldehyde.

**Figure 3 ijms-26-10086-f003:**
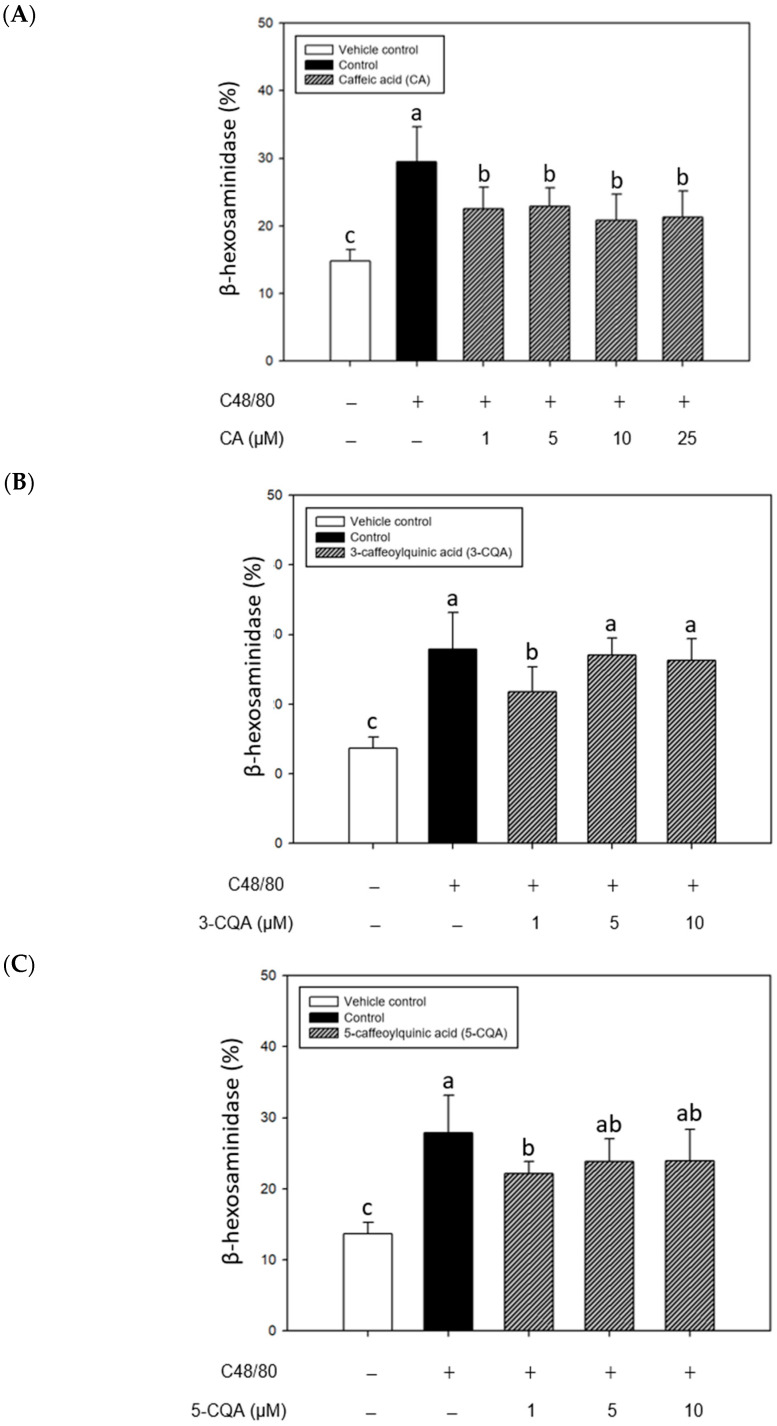
Inhibitory effects of CA (**A**), 3-CQA (**B**), and 5-CQA (**C**) treatments on the β-hexosaminidase release of C48/80-stimulated mouse P815 mast cells. Values are means ± SD (*n* = 6 biological determinations). Bars in the same plot not sharing a common letter (a, b, c) are significantly different (*p* < 0.05) from each other, analyzed by one-way ANOVA, followed by Duncan’s multiple range test. P815 mast cells were sensitized with C48/80 (90 μg/mL). The original cell density was 1 × 10^5^ cells/mL. VC, vehicle control; Control, cell line sensitized with C48/80 without samples; C48/80, compound 48/80. “+”, addition; “−”, without addition.

**Figure 4 ijms-26-10086-f004:**
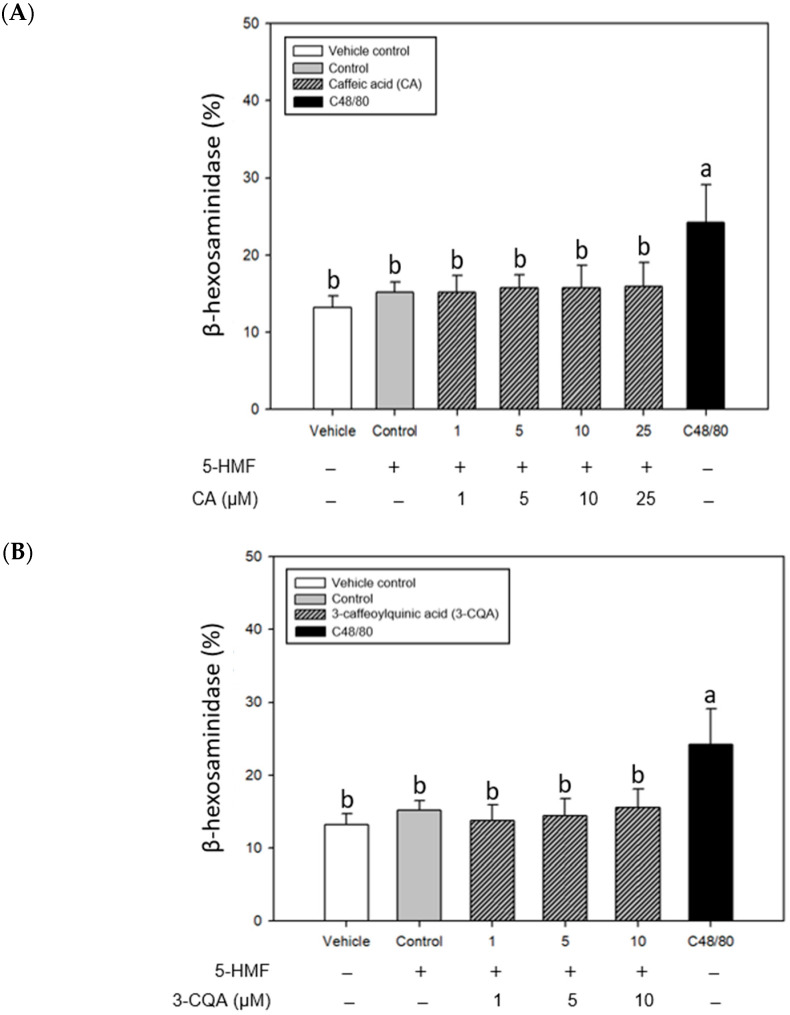
Inhibitory effects of CA (**A**), 3-CQA (**B**), and 5-CQA (**C**) treatments on β-hexosaminidase release of 5-HMF-stimulated mouse P815 mast cells. Values are means ± SD (*n* = 6 biological determinations). Bars in the same plot not sharing a common letter (a, b, c) are significantly different (*p* < 0.05) from each other, analyzed by one-way ANOVA, followed by Duncan’s multiple range test. P815 mast cells were sensitized with 5-HMF (5 mM). The original cell density was 1 × 10^5^ cells/mL. VC, vehicle control; Control, cell line sensitized with 5-HMF without samples; 5-HMF, 5-hydroxymethyl-2-furaldehyde; C48/80, compound 48/80 (90 μg/mL). “+”, addition; “−”, without addition.

**Figure 5 ijms-26-10086-f005:**
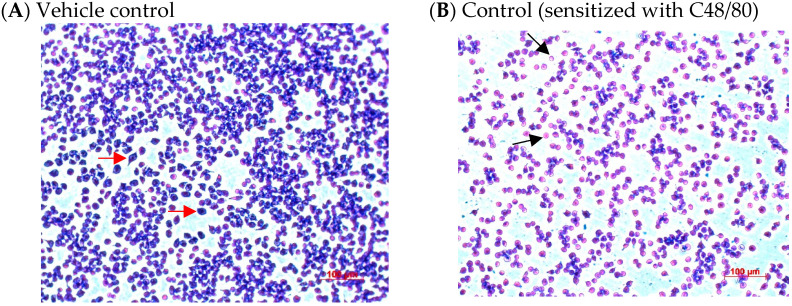
Effects of CA treatments on the appearance and morphology (**A**–**F**), quantification (**G**), and degranulation degrees (**H**) of C48/80-stimulated mouse P815 mast cells, stained with Toluidine Blue. Vehicle control (**A**); Control, cell line stimulated with C48/80 without samples (**B**); Pre-treatment with CA (1, 5, 10, and 25 μM) (**C**–**F**) on C48/80-stimulated P815 mast cells; Quantification of mast cell density (**G**) and degranulation degrees (**H**). Cells with adjacent red arrows indicate regular, and black arrows indicate degranulation. Scale bar = 100 μm. Values are means ± SD (3 random fields). Bars in the same plot not sharing a common letter (a, b, c, d) are significantly different (*p* < 0.05) from each other, analyzed by one-way ANOVA, followed by Duncan’s multiple range test. The cell number of mast cells and degranulation degrees were evaluated using ImageJ’s HSB model (version 1.53). P815 mast cells were stimulated with C48/80 (90 μg/mL). The original cell density was 1 × 10^5^ cells/mL. C48/80, compound 48/80. “+”, addition; “−”, without addition.

**Figure 6 ijms-26-10086-f006:**
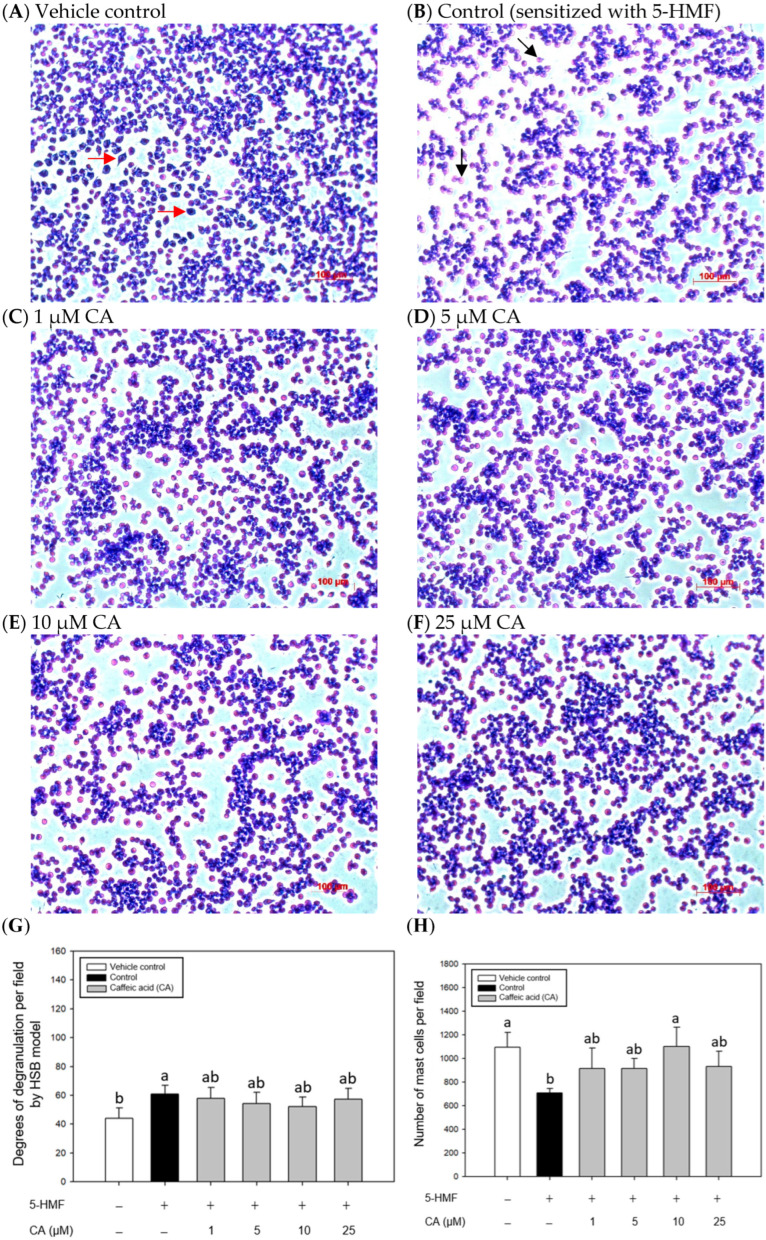
Effects of CA treatments on the appearance and morphology (**A**–**F**), quantification (**G**), and degranulation degrees (**H**) of 5-HMF-stimulated mouse P815 mast cells, stained with Toluidine Blue. Vehicle control (**A**); Control, cells stimulated with 5-HMF without samples (**B**); Pre-treatment with CA (1, 5, 10, and 25 μM) (**C**–**F**) on 5-HMF-stimulated P815 mast cells; Quantification of mast cell density (**G**) and degranulation degrees (**H**). Cells with adjacent red arrows indicate normal, and black arrows indicate degranulation. Scale bar = 100 μm. Values are means ± SD (3 random fields). Bars in the same plot not sharing a common letter (a, b) are significantly different (*p* < 0.05) from each other, analyzed by one-way ANOVA, followed by Duncan’s multiple range test. The cell number of mast cells and degranulation degrees were evaluated using ImageJ’s HSB model (version 1.53). P815 mast cells were sensitized with 5-HMF (5 mM). The original cell density was 1 × 10^5^ cells/mL. 5-HMF, 5-hydroxymethyl-2-furaldehyde. “+”, addition; “−”, without addition.

**Figure 7 ijms-26-10086-f007:**
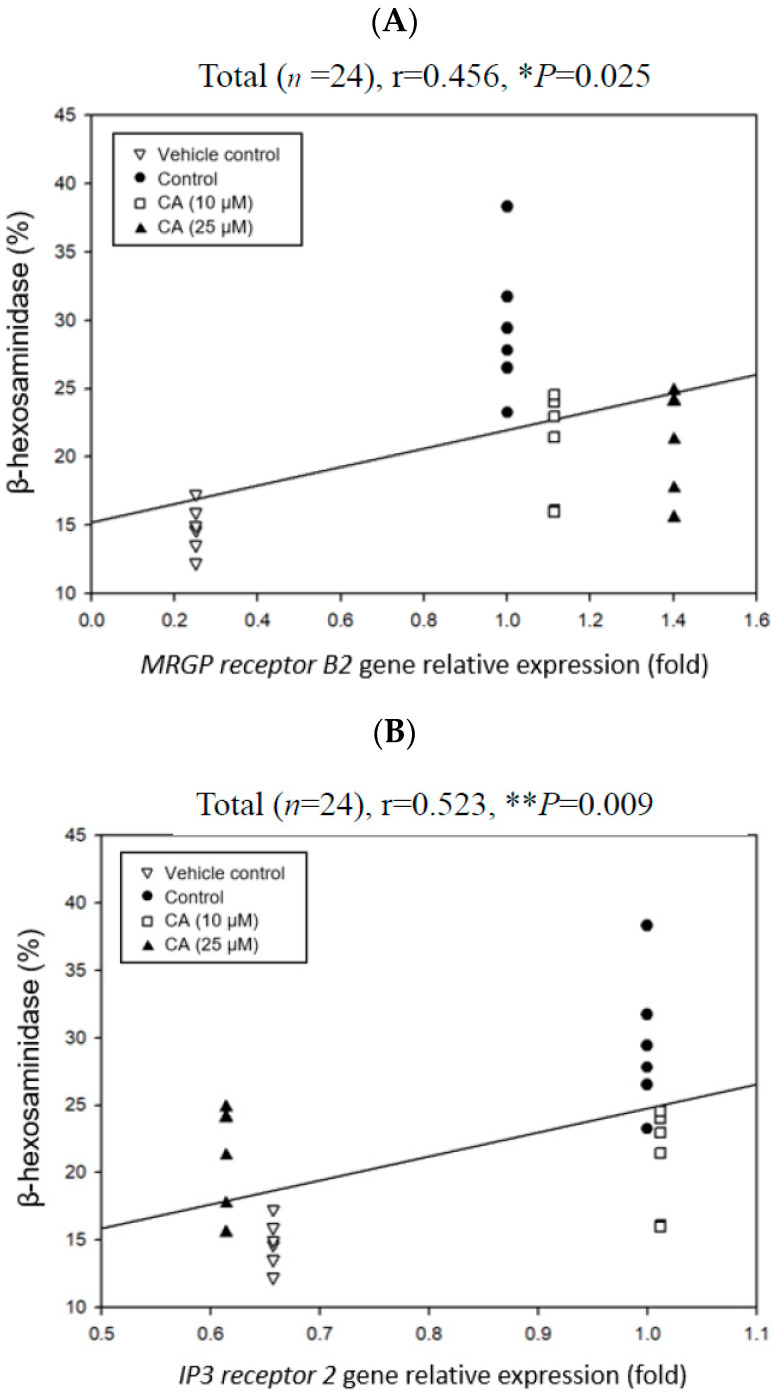
Correlations between relative expression folds of *MRGP receptor B2* (**A**) and *IP3 receptor 2* (**B**) genes involving degranulation-related signaling pathways and the β-hexosaminidase release in mouse P815 mast cells. The correlation was expressed by the Pearson product–moment correlation coefficient (r). * *p* < 0.05; ** *p* < 0.01.

**Figure 8 ijms-26-10086-f008:**
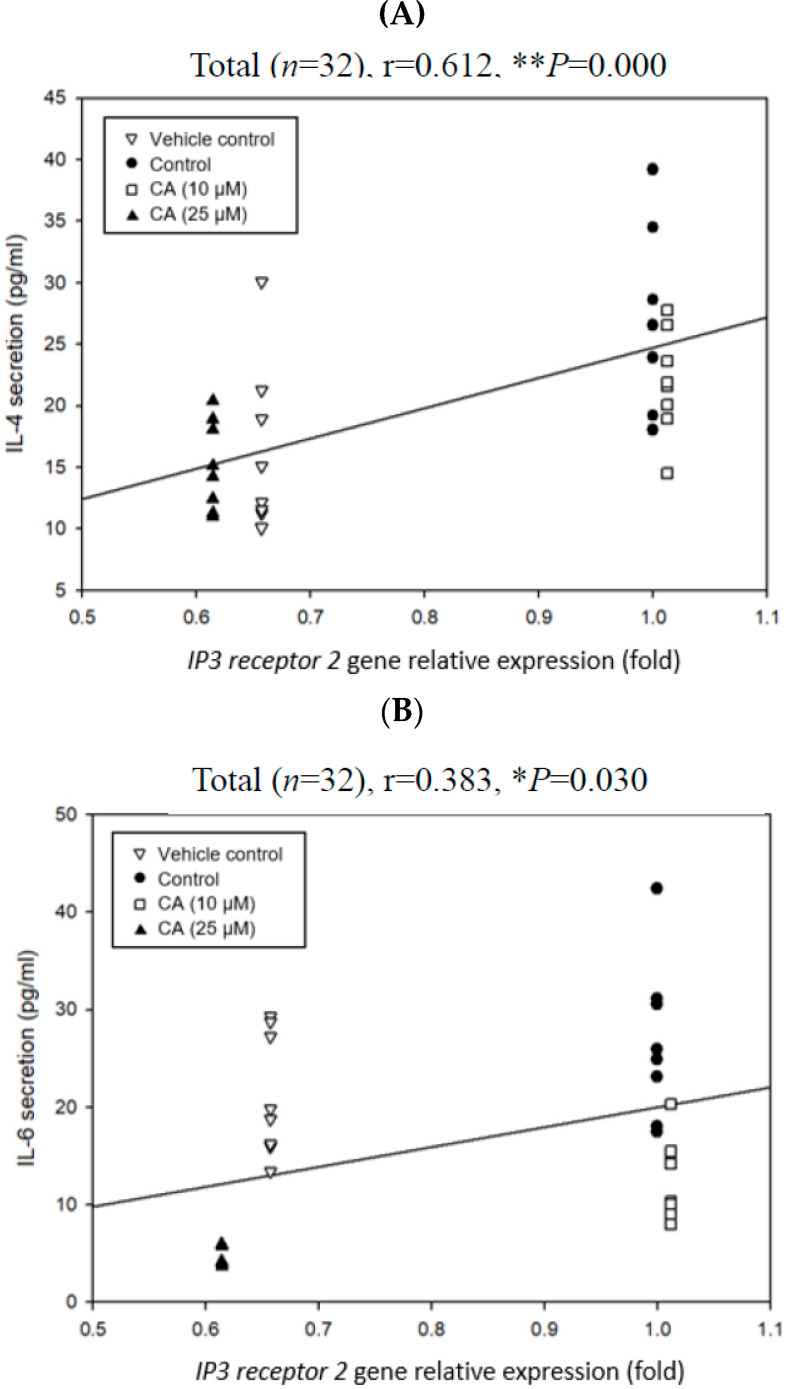
Correlations between relative expression folds of *IP3 receptor 2* gene involving a degranulation-related signaling pathway and IL-4 (**A**) and IL-6 (**B**) secretion levels in mouse P815 mast cells. The correlation was expressed by the Pearson product–moment correlation coefficient (r). * *p* < 0.05; ** *p* < 0.01.

**Figure 9 ijms-26-10086-f009:**
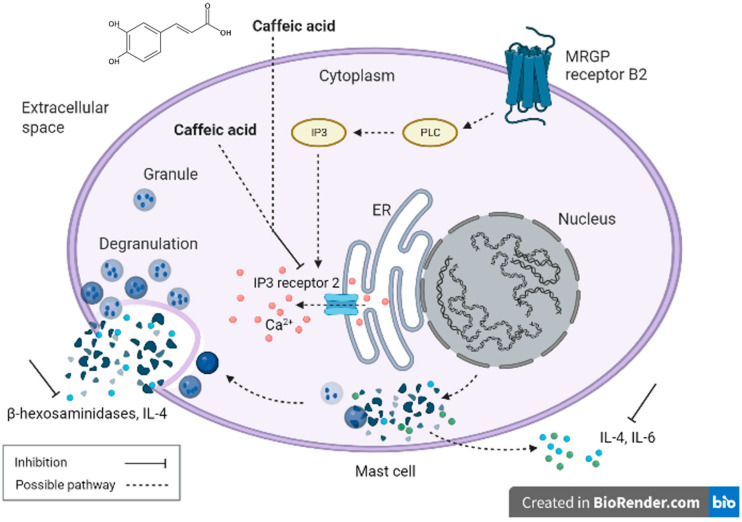
The possible inhibitory mechanisms of the caffeic acid (CA) treatment on degranulation-related signaling pathways in compound 48/80-stimulated mast cells. This experiment unraveled CA’s inhibitory mechanisms on degranulation-related signaling pathways in compound 48/80 (C48/80)-stimulated mast cells. CA might enter the mast cells to exert its immunomodulatory effects by decreasing the gene expression of *IP3 receptor 2*, consequently inhibiting β-hexosaminidase release and the secretion of IL-4 and IL-6 cytokines, thereby slowing down the degranulation of mast cells stimulated by C48/80.

**Table 1 ijms-26-10086-t001:** Effects of CA treatments on secretion levels of IL-4 and IL-6 cytokines by C48/80- and 5-HMF-stimulated mouse P815 mast cells.

Treatments	Conc. (μM)		IL-4 (pg/mL)	IL-6 (pg/mL)
Vehicle control			14.3 ± 4.3 ^c^	21.2 ± 6.3 ^ab^
Control (C48/80 alone)			28.1 ± 7.6 ^ab^	26.7 ± 8.1 ^a^
C48/80 + CA		1	31.6 ± 8.0 ^a^	25.7 ± 9.1 ^a^
		5	27.1 ± 5.7 ^ab^	18.5 ± 6.5 ^bc^
		10	21.9 ± 4.2 ^b^	12.8 ± 4.2 ^c^
		25	15.2 ± 3.6 ^c^	4.6 ± 1.0 ^d^
Vehicle control			14.7 ± 3.8 ^a^	18.0 ± 5.8 ^a^
Control (5-HMF alone)			15.3 ± 8.1 ^a^	5.1 ± 1.7 ^b^
5-HMF + CA		1	15.6 ± 5.0 ^a^	4.3 ± 1.9 ^bc^
		5	12.3 ± 0.9 ^a^	3.6 ± 1.1 ^bc^
		10	13.2 ± 4.9 ^a^	2.8 ± 1.0 ^bc^
		25	11.5 ± 5.1 ^a^	2.1 ± 1.1 ^c^

Values are means ± SD (*n* = 6 biological determinations). Values within the same column under the same treatment item that do not share a common letter (a, b, c, d) are significantly different (*p* < 0.05) from each other, analyzed by one-way ANOVA followed by Duncan’s multiple range test. Cells (1 × 10^5^ cells/mL) were pre-treated with different concentrations of CA for 1 h. ND, not detectable; C48/80, compound 48/80 at 90 μg/mL (Control); 5-HMF, 5-hydroxymethylfurfural at 5 mM (Control). The limit of detection (LOD) of the cytokines ELISA kits used in this study was <15.6 pg/mL.

**Table 2 ijms-26-10086-t002:** Effects of CA treatments on relative expression folds of *MRGP receptor B2* and *IP3 receptor 2* genes in C48/80-stimulated mouse P815 mast cells.

Treatments	Conc. (μM)	Relative Expressions (Fold)	
		*MRGP Receptor B2*	*IP3 Receptor 2*
Vehicle control		0.25±0.10 ^b^	0.66±0.12 ^b^
Control (C48/80 alone)		1.00±0.04 ^a^	1.00±0.03 ^a^
C48/80 + CA	10	1.11±0.04 ^a^	1.01±0.24 ^a^
	25	1.40±0.58 ^a^	0.61±0.03 ^b^

The original cell density was 1 × 10^5^ cells/mL. Values are means ± SD (*n* = 3 biological determinations). Values within the same column not sharing a common letter (a, b) are significantly different (*p* < 0.05) from each other, analyzed by one-way ANOVA, followed by Duncan’s multiple range test. Vehicle control, cell line alone; Control, cell line sensitized with C48/80 without samples; C48/80, compound 48/80; CA, caffeic acid.

## Data Availability

All data are available in the manuscript.

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
