# Peer review of "Caffeic Acid Inhibits Degranulation, Cytokine Secretions, and IP3 Receptor 2 Gene Expression in Compound 48/80-Stimulated Mouse P815 Mast Cells"

_ijms, 2025, doi:10.3390/ijms262010086_

Round 1

Reviewer 1 Report

Comments and Suggestions for Authors

I have to admit that this is an interesting paper. The authors have verified that caffeic acid can have an anti-allergic effect. However, I believe the depth of the research is still insufficient. The potential biological activities and mechanisms of this type of compound have not been fully explored. Of course to help the authors in their revision, I offer the following suggestions.

Author Response

Reviewer 1:

I have to admit that this is an interesting paper. The authors have verified that caffeic acid can have an anti-allergic effect. However, I believe the depth of the research is still insufficient. The potential biological activities and mechanisms of this type of compound have not been fully explored. Of course to help the authors in their revision, I offer the following suggestions.

Response: We thank the Reviewer for the comments. Response to the Reviewer’s comments, together with the itemized changes, has been made in the revised manuscript. The English writing has been improved in the revised manuscript.

  1. After conducting a search of the literature, we found that there are already numerous studies on the anti-allergic effects of caffeic acid. The authors should highlight the innovative aspects of their research.

Response: Page 47 (revised manuscript): We thank the Reviewer for the comments. Two references concerning the anti-allergic effects of caffeic acid in vivo have highlighted the innovative aspects of our research. The description “Although CA has been found to inhibit compound 48/80 and OVA-induced allergic symptoms in allergic dermatitis and allergic asthma in vivo [32,33], this experiment further unravels the anti-allergic mechanisms that CA treatment may alleviate β-hexosaminidase release, IL-4 and IL-6 cytokine secretions, and IP3 receptor 2 gene expression, thereby reducing the degrees of degranulation and inflammation status in C48/80-stimulated mouse P815 mast cells.” has been added to the revised manuscript.

  1. I carefully read the abstract and the introduction, but still couldn't understand why the author chose to study CA, 3-CQA, and 5-CQA. Why not focus solely on CA and complete the research on it?

Response: Page 2 (revised manuscript): We thank the Reviewer for the comments. The description “In our preliminary study, caffeic acid (CA), 3-O-caffeoylquinic acid (3-CQA), and 5-O-caffeoylquinic acid (5-CQA) were suggested to have antiallergic potential among selected CA derivatives. However, inhibitory actions and mechanisms of CA, 3-CQA, and 5-CQA against mast cell degranulation remain unknown. To further unravel the puzzle, CA, 3-CQA, and 5-CQA were subjected to treating stimulated mouse P815 mast cells in vitro in the present study.” has been added to the abstract in the revised manuscript.

        Page 6 (revised manuscript): We thank the Reviewer for the comments. The description “In our preliminary study, CA, 3-O-caffeoylquinic acid (3-CQA), and 5-O-caffeoylquinic acid (5-CQA) were suggested to have antiallergic potential among selected CA derivatives. However, the antiallergic actions and mechanisms of CA, 3-CQA, and 5-CQA remain unclear. Thus, antiallergic effects of CA, 3-CQA, and 5-CQA were explored to determine whether they inhibit the mast cell activation on mouse P815 mast cell degranulation by globally assessing the release of β-hexosaminidase, the appearance and morphological changes, and cytokine secretions.” has been added to the introduction in the revised manuscript.

  1. Since you have chosen to submit your work to this magazine, why don't you make the necessary revisions according to the magazine's formatting requirements? The figures and tables in the result should not be separated.

Response: We thank the Reviewer for the comments. The formatting of the revised manuscript has been updated according to the magazine’s formatting requirements.

  1. The image quality needs to be improved; it is not clear.

Response: We thank the Reviewer for the comments. The image quality has been improved in the revised manuscript.

5. The cell growth curve does not need to be included as a result within it. Since this is something that must be verified in all in vitro cell experiments, what significance does this part of the result have for your conclusion?

Response: Page 9 (revised manuscript): We thank the Reviewer for the comments. To be more concise, the growth curve of the mouse P815 mast cells has been deleted in the revised manuscript. The description “In our preliminary study, the doubling time of the P815 mast cell line was estimated at 23.5 ± 3.1 h, suggesting that the most suitable culture time falls in 24Ë—48 h.” has been added in the revised manuscript.

         Page 16 (revised manuscript): To be more concise, Table 1 in the original manuscript has been deleted in the revised manuscript. However, the description “The primers of the target gene-specific PCR were designed as follows: MRGP receptor B2 (forward primer, FW, 5’→3’): CCCTGGTTGGGATGGGACTA, MRGP receptor B2 (reverse primer, RV, 5’→3’): GAAAGTCAGCCATAGCCAGGT; IP3 receptor 2 (FW, 5’→3’): CCTGACGGTGAACAAGAGGTT, IP3 receptor 2 (RV, 5’→3’): CAGCATTCACATTATCACCCTCG; β-actin (FW, 5’→3’): GCTACAGCTTCACCACCACA, β-actin (RV, 5’→3’): AAGGAAGGCTGGAAAAGAGC.” has been added in the revised manuscript.

  1. The article explicitly uses DMSO to dissolve the drugs. In my opinion, when conducting the MTT experiment, a DMSO control group should be set up.

Response: Page 7 (revised manuscript): We thank the Reviewer for the comments. The description “The stock solutions were sterilized with a 0.20 μm filter (Sartorius, Minisart NY, 17845-ACK, Goettingen, Germany) and diluted with a large amount of Culture medium (CM) or Tyrode’s buffer to reduce DMSO (<1%) residue in the medium when used to eliminate the possible cytotoxicity of DMSO [14,15]. The preliminary studies have reported that DMSO concentration lower than 1% in the medium did not induce any apparent cytotoxicity [14-16]. Since the maximal remaining DMSO concentrations were lower than 1% in the medium in the experiment, a DMSO control group was not included in the cytotoxic experiment.” has been added to the revised manuscript.

  1. The manuscript has no line numbers, so I am unable to specify the problem precisely. The method of constructing the model requires the citation of references.

Response: Pages 7-8 (revised manuscript): We thank the Reviewer for the comments. Four additional references (No. 16-19) concerning the method of constructing the P815 cell culture model have been added to the revised manuscript.

Reviewer 2 Report

Comments and Suggestions for Authors

The Lin and coworker reported the inhibitory action of caffeic acid degranulation, cytokine secretions, and IP3 receptor 2 gene expression in compound 48/80-stimulated mouse P815 mast cells. In this study author reported caffeic acid are more effective than 3-CQA and 5-CQA in inhibiting the β-hexosaminidase release, IL-4 and IL-6 cytokine secretions, and IP3 receptor 2 gene expression in C48/80-stimulated
mouse P815 mast cells. Due to inhibitory effects of caffeic acid on mast cell help to reduce allergic inflammation so it can become a therapeutic agent for allergic diseases. The studies are well executed, well cited and provides meaningful insights into the anti-allergic potential of caffeic acid. I support its for publication

For better understanding author need to cite the references properly. 

Author Response

The Lin and coworker reported the inhibitory action of caffeic acid degranulation, cytokine secretions, and IP3 receptor 2 gene expression in compound 48/80-stimulated mouse P815 mast cells. In this study author reported caffeic acid are more effective than 3-CQA and 5-CQA in inhibiting the β-hexosaminidase release, IL-4 and IL-6 cytokine secretions, and IP3 receptor 2 gene expression in C48/80-stimulated mouse P815 mast cells. Due to inhibitory effects of caffeic acid on mast cell help to reduce allergic inflammation so it can become a therapeutic agent for allergic diseases. The studies are well executed, well cited and provides meaningful insights into the anti-allergic potential of caffeic acid. I support its for publication

For better understanding author need to cite the references properly. 

Response: We thank the Reviewer for the comments. Response to the Reviewer’s comments, together with the itemized changes, has been made in the revised manuscript. Five cited references that belong to our co-authors and 2 other cited references have been removed, but four additional references have been added in the revised manuscript. The English writing has been improved in the revised manuscript.

Round 2

Reviewer 1 Report

Comments and Suggestions for Authors

Thank you very much for your revisions, but there has been no substantive change. I strongly recommend highlighting the innovative aspects. In my opinion, examining certain relevant indicators and genes is not the innovative point of the research; the innovative aspect lies in what has been discovered. I suggest that the author read more articles published in IJMS and make careful revisions.

Author Response

Thank you very much for your revisions, but there has been no substantive change. I strongly recommend highlighting the innovative aspects. In my opinion, examining certain relevant indicators and genes is not the innovative point of the research; the innovative aspect lies in what has been discovered. I suggest that the author read more articles published in IJMS and make careful revisions.

Response: Page 35 (revised manuscript): We thank the Reviewer for the comments. We have cited 5 additional references and the description “CA has been found to possess diverse positive biological potential, particularly in neuroprotective and anti-inflammatory effects, anti-diabetic, anti-atherosclerotic, anti-cancer, and anti-Alzheimer’s diseases [36-40]. In terms of anti-allergic effects, CA has been relatively less studied.” has been added to the revised manuscript.

         Additionally, Table 2 has been improved in the revised manuscript. Each Figure (PDF file type) is included in the revised manuscript.
